# The Relationship between Health Literacy and COVID-19 Vaccination Prevalence during a Rapidly Evolving Pandemic and Infodemic

**DOI:** 10.3390/vaccines10121989

**Published:** 2022-11-23

**Authors:** Iris Feinberg, Jane Yoon Scott, David P. Holland, Rodney Lyn, Lia C. Scott, Kevin M. Maloney, Richard Rothenberg

**Affiliations:** 1Adult Literacy Research Center, College of Education and Human Development, Georgia State University, 30 Pryor Street SW, Suite 750, Atlanta, GA 30303, USA; 2Division of Infectious Diseases, Emory University School of Medicine, 100 Woodruff Circle, Atlanta, GA 30322, USA; 3School of Public Health, Georgia State University, 140 Decatur Street SE, Atlanta, GA 30303, USA

**Keywords:** health literacy, COVID-19 vaccine prevalence, infodemic, vaccine hesitancy

## Abstract

The gap between how health information is communicated and what people understand and can use to make informed health decisions is called health literacy. This gap was exacerbated by the rapidly changing and excessive volume of information, misinformation, and disinformation during the COVID-19 pandemic. People with lower health literacy may not have understood the importance of COVID-19 vaccination for themselves or for their communities. Our aim was to understand health literacy levels within Fulton County, Georgia, and their relationship to vaccine prevalence. Fulton county residents ages 18 and over (*n* = 425) completed an on-line Health Literacy Questionnaire. Individual, organizational, functional, interactive, and critical health literacy scales were created. Vaccination prevalence data were collected from the Georgia Vaccine Distribution Dashboard. All data were divided into one of three county areas. There were statistically significant variations in vaccine prevalence χ^2^(3) = 29.325, *p* < 0.001 among the three county areas. All levels of health literacy predicted overall county vaccination prevalence F (4,420) = 85.941, *p* < 0.001, There were significant differences in health literacy levels among two of the three county area pairs; the lowest resourced county area had the lowest vaccination prevalence and health literacy rates. This is the first example of relating direct health literacy measures across a major metropolitan US county with vaccine prevalence data.

## 1. Introduction

The COVID-19 pandemic and the associated excessive spread of health information, misinformation, and disinformation about the disease, protective measures, and vaccines highlighted the health literacy gap between how messages are communicated and what people can understand and use to make informed health decisions [1,2]. Individual core principles such as choice, freedom, and religious belief have contributed to vaccine refusal, but for many, the plethora of information and misinformation repeatedly delivered through multiple information channels created hesitancy to receive the COVID-19 vaccine [3]. Multiple contradictory narratives increased the reluctance of those who already had concerns about the potential side effects of the vaccination [4]. Narratives that were widely distributed included vaccine-induced infertility, disruption of the immune system, alteration of DNA, and autism [5,6]. Vaccine hesitancy, a disinclination to receive a vaccine regardless of its availability, is a behavior that has been exhibited since the development of vaccine technology and has been correlated with misinformation [6].

Health literacy is the gap between the information that individuals understand and use to make health decisions and how that information is provided. Individual health literacy is multi-dimensional and complex, has evolved over the years, and is difficult to measure [7,8]. Combining different assets and skills, and variable depending on situation and context, health literacy can be measured using many different individual constructs (e.g., reading, numeracy, disease-specific, self-reported) [9], making it problematic to capture, measure, and compare across studies and populations [10]. However, despite differences in process and product, all these measures indicate that to have lower health literacy means not having a range of skills to understand and use health information and services, especially in a rapidly changing environment like the COVID-19 pandemic and infodemic. People around the globe with low health literacy struggled to acquire, understand, and use health information during the COVID-19 pandemic [2,11,12].

It is important to know how people use health literacy skills to navigate the real world of health care, including understanding information, misinformation, and disinformation about the COVID-19 vaccination. The Health Literacy Questionnaire (HLQ) is a validated patient-reported outcomes measure across nine individual construct scales that can be combined to also measure functional, interactive, and critical health literacy skills [7,13]. The HLQ is used across countries, population settings, and economies; results have been used to guide intervention development, communication practices, and health program evaluation [10]. In particular, findings can help healthcare organizations and providers of health information like public health agencies recognize how people experience the healthcare system, including understanding and using health information [14,15]. In this study, we used the HLQ to understand the relationship between lived health literacy experiences and COVID-19 vaccine prevalence across Fulton County, the largest county in Georgia, which is composed of three distinct geographical county areas: North Fulton, the city of Atlanta, and South Fulton. Our research questions are:Is there a difference among county areas (North Fulton, South Fulton, Atlanta) in vaccination prevalence and in demographic factors that might predict vaccination prevalence?How do individual and organizational health literacy scores for Fulton County residents vary by county area (North Fulton, South Fulton, Atlanta)?Is functional, interactive, and critical health literacy associated with vaccination prevalence by county area?

## 2. Methods

### 2.1. Sample

People ages 18 and older who live in Fulton County were recruited using email, text, posted flyers, and live recruitment in May, June, and July 2022. Invitations to participate were emailed and texted to individuals who live in Fulton County who had participated in the Principal Investigator’s prior health literacy studies between 2015–2020 and emailed to all employees of the Fulton County Board of Health. Flyers with QR codes were posted in Fulton County libraries and public health facilities, and live recruitment occurred at the Fulton County Board of Health Clinic in downtown Atlanta. Interested respondents accessed a Qualtrics survey through a hyperlink. Participants were paid a $25 Visa or Amazon gift card upon completion of the survey. The study was approved by Georgia State University’s Institutional Review Board.

### 2.2. Measures

Data were collected on age, sex, race, health insurance status, educational status, and zip code. County location (North Fulton, Atlanta, South Fulton) was determined using visual mapping and postal zip code data. If a zip code crossed into more than one county location or into another county entirely, we evaluated where the bulk of the zip code fit. Overall, 95% of the zip codes aligned with one of the three county locations.

The HLQ was used to collect health literacy information; responses to the original 44 questions create nine scales with a high overall reliability scale of >0.08 [10]. The nine scales contain four to six items scored on a Likert-style scale; four scales have four response options (strongly agree, agree, disagree, strongly disagree) and five scales have five response options (cannot do, very difficult, difficult, easy, very easy). Seven of the scales represent individual health literacy domains: having sufficient information to manage my health, actively managing my health, social support for health appraisal of health information, ability to actively engage with healthcare providers, ability to find good health information, and understand health information enough to know what to do. Two of the scales represent organizational health literacy domains: feeling understood and supported by a healthcare provider and navigating the healthcare system. There is no overall total score, and no standard score for each scale to indicate low health literacy [10,16]. We followed methods used by other researchers to create binary variables to measure lower and higher health literacy [10,14,15,17]. For scales with scores of 1–4, the cut off was 50%; for scales with scores of 1–5, the cut off was less than or equal to 60 [17].

We created three z-scored health literacy composite scales (functional, interactive, critical). Information on the original HLQ scales and composite scales are in Appendix A.

We assessed COVID-19 vaccine prevalence on 17 July 2022, using the Georgia Vaccine Distribution Dashboard, Vaccination Prevalence by Census Tract report [18]. Data were provided for first vaccination and complete vaccination (2 shots) for people ages 6 months and older; we used the complete vaccination data. Children ages 6 months to 19 years old were 12.6% of the total pool. We cross-walked the census tracts to zip codes using the Housing and Urban Development Census Tract to Zip Code Crosswalk report [19]. For participants, we received zip code data and geocoded them to North or South Fulton or City of Atlanta. Vaccination prevalence by county area was calculated by weighting the number of vaccines given by the population count in each zip code.

We used SPSSv.27 for analysis (IBM Corp., Armonk, NY, USA, 2020). Descriptive statistics included means and standard deviations (sd) for continuous variables, and frequencies and percentages for categorical variables. We utilized the Kruskal–Wallis H test to determine differences between two or more groups of rank-based nonparametric data and used a Bonferroni correction for multiple comparisons. We used multiple linear regression analyses to predict demographic variables and functional, interactive, and critical health literacy to vaccine completion prevalence at the population level.

## 3. Results

We recruited 519 participants; the final survey sample was 425 people (81.9%). Participants were deemed ineligible if they were not a Fulton County resident, their response times were greater than two standard deviations less than the mean of average time to complete, and their responses indicated straight lining. The mean age of the Fulton County sample was 40.8 years (sd 13.9), with a range of ages from 18–78. A large majority of participants were African American or Black (85%); 72% were women. Using zip code and census tract data, we organized participants into three areas: North Fulton (21.2%), Atlanta (34.6%), and South Fulton (44.2%). Vaccination prevalence ranged from 50.7% (South Fulton) to 71.9% (North Fulton), and 65.7% of Atlanta residents were fully vaccinated. The overall Fulton County vaccination prevalence was 62.4%. See Table 1.

To answer research question one, “Is there a difference among county areas (North Fulton, South Fulton, Atlanta) in vaccination prevalence and in demographic factors that might predict vaccination prevalence?”, we assessed overall variance as well as pairwise comparisons of the three county areas. Overall, vaccine prevalence varied among the three county areas (χ^2^(3) = 29.325, *p* < 0.001). Pairwise comparisons revealed statistically significant differences in complete vaccination prevalence between all of the county areas (*p* < 0.001). We looked at predictions for vaccinations to see if age, sex, educational attainment, or health insurance status predicted vaccination prevalence. For complete vaccination, only age predicted vaccination status, F(4, 420) = 5.179, *p* = 0.000; as age decreased, complete vaccination status increased. See Table 2.

To answer research question two, “How do individual and organizational health literacy scores for Fulton County residents vary by county area (North Fulton, South Fulton, Atlanta)?”, individual and organizational health literacy scales are reported in Figure 1, Figure 2, Figure 3 and Figure 4. South Fulton participants had the lowest health literacy across all domains.

To answer research question three, “Is functional, interactive, and critical health literacy related to vaccination prevalence by county area (North Fulton, South Fulton, Atlanta)?”, we looked at predictions of functional, interactive, and critical health literacy to vaccination prevalence controlling for age (the only significantly contributing variable; see research question one). Three z-scored health literacy composite scales (functional, interactive, critical) were created.

R^2^ for the overall model was 0.450 with an adjusted R^2^ of 44.5%, a medium-sized effect according to Cohen (1988). Controlling for age, the multiple regression model statistically significantly predicted Complete Vaccination prevalence F (4,420) = 85.941, *p* < 0.001 with all variables significant. See Table 3.

When reviewing pairwise comparisons with all three county areas in the Kruskal–Wallis test, we found no statistically significant difference between North Fulton and Atlanta in any health literacy category. However, we found significant differences between South Fulton and both of the other county areas, with South Fulton respondents scoring lower in all three health literacy domains. See Appendix A for results of pairwise comparisons and Figure 5 for county area estimates of vaccination prevalence and health literacy levels (functional, interactive, critical).

## 4. Discussion

The intent of this study was to understand the relationship between health literacy and COVID-19 vaccination prevalence during a rapidly evolving pandemic with massive amounts of COVID-19 information, misinformation, and disinformation flooding the airwaves, print sources, and the internet. We found that county areas with higher health literacy levels had higher vaccination prevalence than areas with lower health literacy. Our study also shows that age is related to COVID-19 vaccine uptake. Controlling for age, health literacy levels appear to predict higher COVID-19 vaccination prevalence in Fulton County.

During the second and third waves of the coronavirus pandemic, researchers sought to better understand people’s intentions for vaccine uptake. Globally, there are only a few studies where health literacy and vaccine intention were measured, and to the best of our knowledge, no study other than ours studies the relationship between health literacy and vaccine prevalence. For example, a systematic review conducted via MEDLINE and EMBASE studying the channels of communication, source credibility, and how health risk messages are communicated pertaining to health risk found that the readability of government and health websites pertaining to COVID-19 vaccines were often too sophisticated for the public to understand [20]. Turkish researchers affirmed the relationship between vaccine hesitancy and health literacy; people with difficulty understanding COVID-19 vaccine information, i.e., those with lower health literacy, were more likely to experience vaccine hesitancy [21]. Other researchers showed that the ability to detect fake news and health literacy scores were associated with COVID-19 vaccine intention; the risk of poor vaccine intention was higher among those having lower health literacy (OR = 1.44; 95%CI = 1.04, 2.00) [22]. Studies in Italy show the relationship between difficulties in accessing and understanding vaccine information and lower health literacy among a cross-sectional survey of 3500 participants [23]. Most studies assessed self-reported vaccine hesitancy; according to Hu et al., in a study assessing vaccine hesitancy and vaccination prevalence across the US, there is a large discrepancy between self-reported vaccine hesitancy and actual vaccination coverage (2022) [24]. Due to the availability of completed administrative vaccination data from the Georgia Department of Public Health, we were able to directly measure heath literacy rates and compare them with actual vaccine compliance.

Structural inequalities also exacerbate low individual health literacy; low COVID-19 knowledge, poor COVID-19 protective and vaccination behaviors, and poor health outcomes are associated with social determinants of health that are also related to low health literacy [25,26,27]. People with lower health literacy face innumerable challenges assessing and using health information in general. During the COVID-19 pandemic, an inability to engage with and act upon rapidly evolving critical health information about mitigation strategies and the need for vaccination deeply affected people with lower health literacy [28]. From a public health perspective, greater individual health literacy skills can lead to greater personal empowerment, knowledge, and agency, which may ultimately lead to informed decisions and actions that can improve health outcomes. Strong organizational health literacy efforts can help individuals meet the complex demands of the health environment.

People need trustworthy and accurate information that they understand to make the most informed decisions about health and well-being during epidemics, pandemics, health crises, and everyday life. During the pandemic, it was impossible for people to parse the plethora of rapidly changing COVID-19 messages to understand what behaviors and actions would be the most protective. An investment in organizational health literacy to ensure effective and trustworthy understandable health information could be one of the most effective strategies in the battle against misinformation and disinformation. The long-term perspective of individual health literacy as an asset, not an individual deficit, can be bolstered by organizational, structural, and systematic actions that pay attention to vulnerable and higher-risk communities [1].

There are common organizational elements that can be implemented to reduce the health literacy demand on patients and health consumers and may lead to improved patient outcomes. One example is to establish a workforce that has organizational health literacy skills and knowledge; for example, evidence-based practices such as Teach-Back can be part of professional development. Teach-Back is an easily learned method for confirming patient understanding [29,30,31], which can be measured at the organizational level (e.g., how many staff have been trained, number of electronic health record/charts that indicate Teach-Back was used) and at the patient level (e.g., electronic health record/charts that indicate patient correctly taught back medical instructions; patient following medication and discharge instructions appropriately, including follow-up appointments). Improving access to written and web-based patient education developed using plain language and health literacy guidelines is a second example of how organizations can become more health literate. One in five US adults reads at elementary levels [32]; documents and web-based educational materials should not only be readable at the 8th grade level, but should also be written in plain language that ensures people cannot only read the words, but can understand what they read the first time [33]. National culturally and linguistically appropriate standards (CLAS) should be implemented to ensure equitable, understandable, and respectful access to health care for people who are non-native English speakers [34]. Federal regulations require access to interpretive services for non-native English speakers, as well as appropriate navigation signage. Written materials should not only be translated, but should also go through a continuous review process to ensure they are also culturally responsive for the intended audience.

While improving organizational health literacy may not tackle the problem of misinformation, disinformation, or just too much information, providing people with information that they can at least understand and use to make informed decisions is a credible first step in narrowing the health literacy gap. The COVID-19 pandemic and infodemic clearly showed that people whose social and demographic characteristics are more often correlated with lower health literacy suffered inequitably [1,5,18]; improving health literacy may contribute to reducing disparities in health outcomes.

### Limitations

The results of this study should be interpreted in the context of a few limitations. While our sample is broadly representative of Fulton County residents, it does not mirror the demographics of the county, including digital access and digital skills. Although we made the survey available in Spanish, only English-speaking people participated. Some data were only available in census tract format, and other data were only available in zip code format; we used the most widely accepted census tract-zip code crosswalk but were not able to match all data. Zip codes and census tracts also often cross city, county area, and county lines; therefore, in those cases we had to use best judgement and visual mapping to determine location. This cross-sectional survey was ecological in nature, thus the prevalence of vaccination estimated for the county areas may not reflect vaccine uptake among our survey respondents. Finally, these data represent survey results and vaccination prevalence during a specific time during the COVID-19 pandemic (July, 2022). We may have seen lower vaccination prevalence if completing the study earlier in the pandemic.

## Figures and Tables

**Figure 1 vaccines-10-01989-f001:**
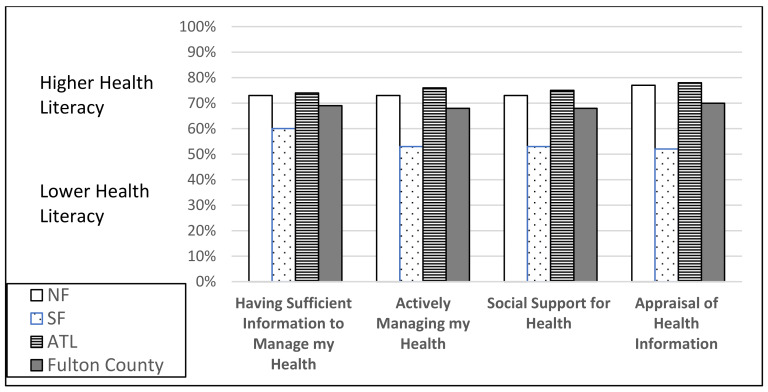
Individual Health Literacy Scale (cut point 50%) by County Area: Sufficient Information, Managing Health, Social Support, Appraisal of Information. Based on Level of Agreement (lower = strongly disagree/disagree; higher = agree/strongly agree).

**Figure 2 vaccines-10-01989-f002:**
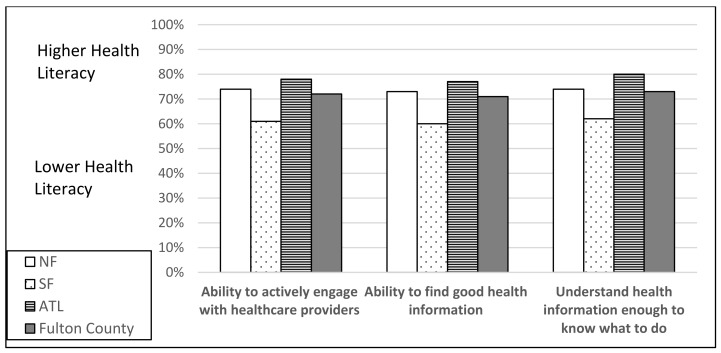
Individual Health Literacy Scale (cut point 60%) by County Area: Engage with Providers, Find Health Information, Understand Health Information. Based on Level of Difficulty (lower = cannot do/very difficult/quite difficult; higher = easy/quite easy).

**Figure 3 vaccines-10-01989-f003:**
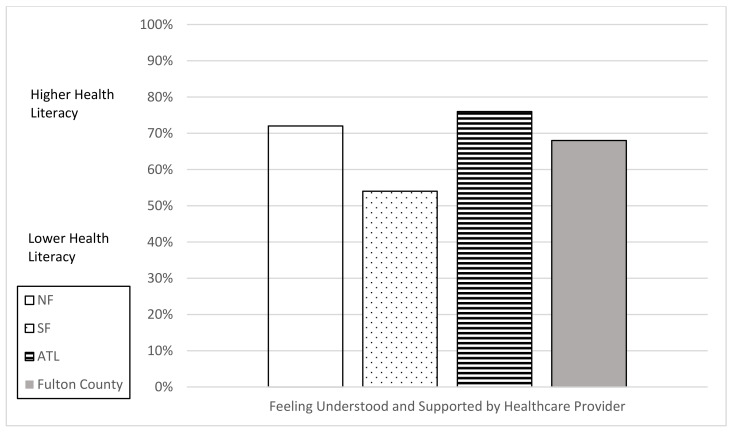
Organizational Health Literacy Scale (cut point 50%) by County Area: Understood and Supported by Provider. Based on Level of Agreement (lower = strongly disagree/disagree; higher = agree/strongly agree).

**Figure 4 vaccines-10-01989-f004:**
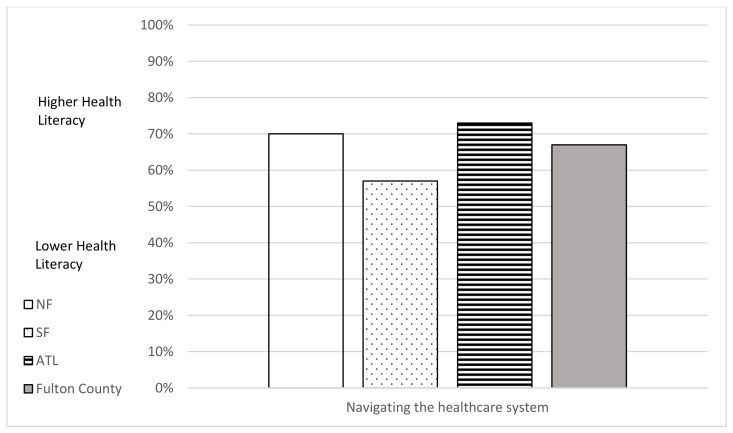
Organizational Health Literacy Scale (cut point 60%) by County Area: Navigating the Healthcare Systems. Based on Level of Difficulty (lower = cannot do/very difficult/quite difficult; higher = easy/quite easy).

**Figure 5 vaccines-10-01989-f005:**
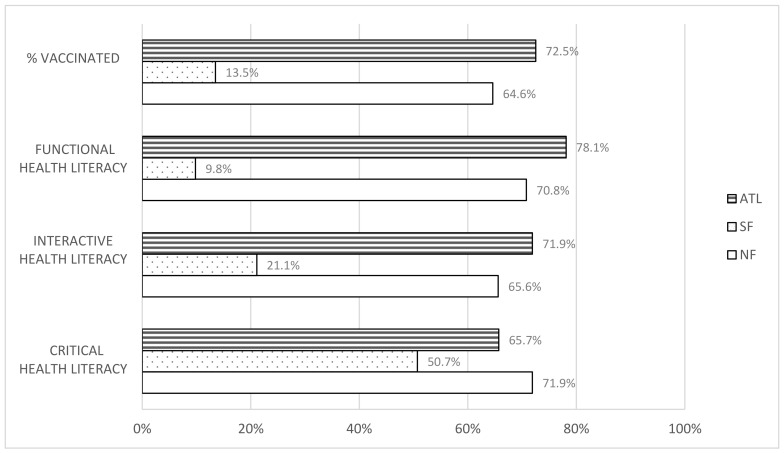
Complete Vaccination Prevalence and Health Literacy Levels (Functional, Interactive, Critical) by County Area. %VACCINATED = Complete Vaccination Prevalence; NF = North Fulton, SF = South Fulton, ATL = Atlanta.

**Table 1 vaccines-10-01989-t001:** Demographics and Vaccination Prevalence by County Area.

	North Fulton (21.2%)			South Fulton (44.2%)			Atlanta (34.6%)			Total Fulton County		
	*n*	%	Mean (Sd)	*n*	%	Mean (Sd)	*n*	%	Mean (Sd)	*n*	%	Mean (Sd)
Age	90		36.7 (12.9)	188		44.7 (14.3)	147		40.1 (13.4)	425		40.8 (13.9)
Female	60	66.7%		144	76.6%		101	68.7%		305	71.8%	
High School Diploma or Less	15	16.7%		28	14.9%		17	11.6%		60	14.1%	
No Health Insurance	16	17.8%		25	13.3%		24	16.3%		65	15.3%	
African American or Black	76	84.4%		166	88.3%		129	87.8%		371	87.3%	
Complete Vaccination Prevalence *		71.9%			50.7%			65.7%			62.4%	

* Estimated at the population level with an external data source, not among survey participants.

**Table 2 vaccines-10-01989-t002:** Multiple Linear Regression Results for Complete Vaccination Prevalence.

Variable	*B*	*SE B*	Beta	*t*	*p*
Gender	0.61	0.863	0.035	0.707	0.48
Age	−0.125	0.029	−0.21	−4.265	0.00 *
Health Insurance Status	0.276	1.132	0.012	0.243	0.808
Education	0.502	1.16	0.021	0.433	0.665

* Significant at α < 0.05.

**Table 3 vaccines-10-01989-t003:** Multiple Regression Results for Complete Vaccination Prevalence Controlling for Age.

Complete Vaccination	*B*	95% CI for *B*		*SE B*	β	*R* ^2^	Δ*R*^2^
		LL	UL				
Model						0.450	0.445
Constant	67.257	65.410	69.104	0.940		0.450	
Age	−0.119	−0.162	−0.076	0.022	−0.200 **		
FHL *	−1.211	−1.830	−0.591	0.315	−0.395 **		
IHL *	0.718	0.268	1.168	0.229	0.449 **		
CHL *	1.600	1.080	2.119	0.264	0.533 **		

* FHL (Functional Health Literacy) IHL (Interactive Health Literacy) CHL (Critical Health Literacy); ** significant at *p* < 0.001.

## Data Availability

The data presented in this study are available on request from the corresponding author. The data are not publicly available due to privacy.

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
