# Peer review of "The Relationship between Health Literacy and COVID-19 Vaccination Prevalence during a Rapidly Evolving Pandemic and Infodemic"

_vaccines, 2022, doi:10.3390/vaccines10121989_

Round 1

Reviewer 1 Report

Thank you for your submission. While it is quite interesting and well-written, it is not novel and has major scientific issues. Many other studies worldwide have examined the relationship.

The research aim should be stated in the abstract. The abstract should also include how health literacy was assessed.

It is odd that the researchers did not directly ask the participants about their COVID-19 vaccination status, and assess the correlation between individuals’ health literacy and vaccination status. Why look indirectly at population level vaccination status? This is a very imperfect approach, especially when attempting to factor in numerous other variables/potential confounders e.g. age, sex, educational attainment, health insurance status.

Where is the multivariate analysis examining the relationship between HLQ and vaccination status while controlling for potential confounders (e.g. age)?  This must be performed.

When did recruitment and the HLQ assessment occur?

This is a retrospective observational study. Causality cannot be ascertained. All we have are statistical associations. Therefore, statements such as that at the end of the abstract should be refined: “This is the first example of comparing health literacy levels across a major metropolitan county with vaccine prevalence data, showing how people’s health literacy levels may be associated with their willingness to take the COVID-19 vaccine.”

Author Response

Thank you for your insightful comments.   Our responses are below.

  1. Many other studies worldwide have examined the relationship.

I am curious as to which studies you are referring to.  Very few direct measures of health literacy were done during the COVID period.  Further, any correlation reported looked at self-reported vaccine hesitancy, not vaccine prevalence. 

  1. The research aim should be stated in the abstract. The abstract should also include how health literacy was assessed.

We have added this sentence to the abstract:  “Our aim was to understand health literacy levels within Fulton County, Georgia, and their relationship to vaccine prevalence.”

  1. It is odd that the researchers did not directly ask the participants about their COVID-19 vaccination status and assess the correlation between individuals’ health literacy and vaccination status. Why look indirectly at population level vaccination status? This is a very imperfect approach, especially when attempting to factor in numerous other variables/potential confounders e.g. age, sex, educational attainment, health insurance status.

This is a valid criticism, although we already discussed this in the limitations paragraph in the Discussion: “This cross-sectional survey was ecological in nature, so the prevalence of vaccination estimated for the county areas may not reflect vaccine uptake among our survey respondents.”   The initial data collection was to assess health literacy only.  We determined after the fact that it would be an interesting analysis to compare health literacy with vaccine prevalence.

  1. Where is the multivariate analysis examining the relationship between HLQ and vaccination status while controlling for potential confounders (e.g. age)?This must be performed.

The multiple regression analysis was performed, and results reported (see research question 3).  We added a sentence indicating that we controlled for the demographic variables on page 6, line 179. “To answer research question 3, "Is functional, interactive, and critical health literacy related to vaccination prevalence by county area (North Fulton, South Fulton, Atlanta)?”, we looked at predictions of functional, interactive, and critical health literacy to vaccination prevalence controlling for gender, educational attainment, insurance status, and age.”

  1. When did recruitment and the HLQ assessment occur?

We have added the date range on page 2, line 81. “People ages 18 and older who live in Fulton County were recruited using email, text, posted flyers and live recruitment in May, June and July 2022.”

  1. Therefore, statements such as that at the end of the abstract should be refined: “This is the first example of comparing health literacy levels across a major metropolitan county with vaccine prevalence data, showing how people’s health literacy levels may be associated with their willingness to take the COVID-19 vaccine.”  

We have changed this statement to read: “This is the first example of comparing direct health literacy measures across a major metropolitan county with vaccine prevalence data.”

Reviewer 2 Report

This is a well written study and other than the limited number of participants there are no major issues.

May I suggest some ways in which the study can be improved. I believe that there is an index of multiple deprivation available at county and perhaps electoral district level, there may also be other social grouping (i.e., anything similar to commercial consumer groups) data available. Is it possible to correlate the health literacy scores with such measures of deprivation and/or social grouping? This would allow you to group all the respondents across the 3 counties by deprivation/ social group rather than by county.

This would be somewhat akin to the cherry on top of the cake.

Author Response

Thank you for your comments.  We did consider evaluating the Social Vulnerability Index (SVI) as part of our study.  The SVI is only available by census tract, and represents a ranking, not a value.  Thus, we cannot aggregate it, nor can we create SVIs for each of our county areas.  Each of our county geographic areas also has SVI rankings from the lowest to the highest.  It would make a very interesting study to evaluate vaccine prevalence to SVI since both are reported by census tract.  We will consider doing this in future work.

Round 2

Reviewer 1 Report

Thank you for your revision. However, I still believe the study lacks sufficient novelty and scientific rigour for publication in Vaccines.

There are countless studies (and now reviews) examining the association between health literacy and COVID-19 vaccine hesitancy, which we know is very closely correlated with vaccine uptake. In addition, here are some studies that have examined vaccine uptake itself relative to health literacy.

·         Lastrucci V, Lorini C, Stacchini L, et al. Determinants of Actual COVID-19 Vaccine Uptake in a Cohort of Essential Workers: An Area-Based Longitudinal Study in the Province of Prato, Italy. Int J Environ Res Public Health 2022;19(20)

·         Heiniger S, Schliek M, Moser A, von Wyl V, Höglinger M. Differences in COVID-19 vaccination uptake in the first 12 months of vaccine availability in Switzerland - a prospective cohort study. Swiss Med Wkly. 2022 Apr 8;152:w30162. 

·         Pickles K, Copp T, Meyerowitz-Katz G, Dodd RH, Bonner C, Nickel B, Steffens MS, Seale H, Cvejic E, Taba M, Chau B, McCaffery KJ. COVID-19 Vaccine Misperceptions in a Community Sample of Adults Aged 18–49 Years in Australia. Int J Environ Res Public Health 2022; 19(11):6883.

It still seems pointless to indirectly examine vaccination status via inference from location of residence.

The researchers should have directly asked the participants about their COVID-19 vaccination status, and assess the correlation between individuals’ health literacy and vaccination status. Why look indirectly at population level vaccination status? This is a very imperfect and crude scientific approach, especially when attempting to factor in numerous other variables/potential confounders e.g. age, sex, educational attainment, health insurance status.

The multivariate analysis examining the relationship between HLQ and vaccination status while controlling for potential confounders (e.g. age) is still unclear/absent. The method should be clearly explained. Where is a table showing the results (e.g. adjusted and unadjusted odds ratios) for all variables simultaneously?  Table 3 is not this.  Was age a more important predictor than health literacy?

Author Response

Thank you for sharing the vaccine hesitancy literature.  We are aware of these studies and more, and find all of them to be particularly telling about health literacy and the need for improving health communication as it relates to vaccines.

Regarding our choice to evaluate vaccine prevalence rather than self-reported vaccination status, a recent manuscript published in Vaccines establishes that vaccine hesitancy (self-reported in Household Pulse Survey) does not accurately represent actual vaccination rates (reported in CDC COVID Data Tracker) and "...challenges the popular narrative that portrays vaccine hesitance as a root cause of disparities in vaccination".  We agree with the authors that a variety of exogenous factors are responsible for hesitancy and coverage and that more research into this relationship is necessary.  However, as this special call was specifically related to health literacy, we examined the relationship between health literacy and vaccination prevalence.  The Vaccine article can be found here: doi: 10.1016/j.vaccine.2022.07.051

Regarding this comment: The multivariate analysis examining the relationship between HLQ and vaccination status while controlling for potential confounders (e.g. age) is still unclear/absent. The method should be clearly explained. Where is a table showing the results (e.g. adjusted and unadjusted odds ratios) for all variables simultaneously?  Table 3 is not this.  Was age a more important predictor than health literacy?

I apologize - inadvertently submitted the last revision without the correct table.  It is now corrected.  We have corrected the text lines 195-199 as well.